# Evaluation of Women’s Age and Ultrasonographic Features to Choose Surgical Treatment for Endometriosis Associated with Ovarian Cancer

**DOI:** 10.3390/jcm11092414

**Published:** 2022-04-25

**Authors:** Alicia Hernández, Angela Sanz, Emanuela Spagnolo, María Carbonell, Elena Rodríguez, Ana López, Riccardo Raganato, Simona Del Forno, David Ramiro-Cortijo

**Affiliations:** 1Department of Obstetrics and Gynecology, Hospital Universitario La Paz, Paseo de la Castellana, 261, 28046 Madrid, Spain; aliciahernandezg@gmail.com (A.H.); angelasanzmaset@gmail.com (A.S.); mariacarbonell676@gmail.com (M.C.); elenarogon@gmail.com (E.R.); analopezcarrasco.lopez@gmail.com (A.L.); 2Department of Obstetrics and Gynecology, Faculty of Medicine, Universidad Autónoma de Madrid, C/Arzobispo Morcillo 2, 28029 Madrid, Spain; 3Department of Orthopaedic Surgery and Traumatology, Hospital Universitario La Paz, Paseo de la Castellana, 261, 28046 Madrid, Spain; riccardo.raganato@gmail.com; 4Division of Gynecology and Human Reproduction Physiopathology, Department of Medical and Surgical Sciences (DIMEC), IRCCS Azienda Ospedaliero, S. Orsola Hospital, University of Bologna, Via Masserenti 13, 40138 Bologna, Italy; simona.delfo@hotmail.it; 5Department of Physiology, Faculty of Medicine, Universidad Autónoma de Madrid, C/Arzobispo Morcillo 2, 28029 Madrid, Spain; david.ramiro@uam.es

**Keywords:** ovarian cancer, endometriosis-associated ovarian cancer, CA 125, ultrasound, risk factor

## Abstract

Adequate surgical management of malignant endometriosis remains a clinical challenge in gynecology. Age, sonography variables, and tumor biomarkers have been reported as candidates in the clinical decision. This study aims were to analyze the factors of women’s age, body mass index, ultrasound features, and tumor biomarkers to predict endometriosis-associated ovarian cancer in a large series of endometriomas and to study the surgical treatment performed in this cohort. In this retrospective study, we reviewed the medical records of patients with ultrasound diagnosis of ovarian cyst classified as endometrioma (benign as well as with risk of malignancy), surgically treated in the endometriosis unit of Hospital Universitario La Paz (Madrid, Spain) between January 2019 and July 2021. According to the final histology examination, the women were clustered as non-endometriosis-associated ovarian cancer (OE, benign endometriomas, *n* = 59) and endometriosis-associated ovarian cancer (EAOC) (*n* = 17). Demographic, clinical, and surgical data were collected from these women. International Ovarian Tumor Analysis (IOTA) criteria were assessed for the ultrasound examination. The age of the women in the EAOC group was 50.0 [43.0; 63.0] years, which was significantly higher than OE (39.0 [34.0; 46.0] years; *p*-value < 0.001). In addition, the body mass index for the OE group (24.9 ± 5.3 kg/m^2^) was significantly higher than for the EAOC group (23.3 ± 4.6 kg/m^2^; *p*-value < 0.001). However, the tumor biomarker levels (CA 125, CA 19.9 and He4) were not significantly different among the groups. We performed 51.4% cystectomies and 48.6% adnexectomies, with an association between the adnexectomy and EAOC group (*p*-value < 0.001). In addition, a significant association was found between ultrasound features suspicious for malignancy and the EAOC group. Conclusively, women’s age and ultrasound features, such as papillary projections, septa, and positive echo-Doppler, were the main factors to consider when evaluating the malignancy risk associated with endometriosis.

## 1. Introduction

Endometriosis, defined by endometrial-like tissue (stroma and glands) outside the uterine cavity [1], is a common gynecological disease that affects 5–10% of women of reproductive age. The ectopic foci of endometrial tissue distort the pelvic anatomy through inflammatory reaction, adhesions, and scar tissue, hence, causing dysmenorrhea, dyspareunia, chronic pelvic pain, and infertility [2]. Despite decades of research, the pathogenesis of endometriosis is not completely understood, with multiple theories postulated regarding its etiology [3]. In addition, hormonal, immunological and genetical pathways have been suggested to play a role in the development of endometriosis [4].

Although endometriosis is thought of as a mostly benign gynecological disease, data indicate that endometriosis is associated with a significant risk for developing specific forms ovarian carcinoma [5]. The most common types are endometrioid and clear-cell subtypes. They are usually diagnosed in an early stage and affect young women of reproductive age. Histopathological, epidemiological, and molecular findings indicate that endometriosis has malignant potential [6]. The incidence of malignant transformation of endometriosis has been described to be 0.7–2.5% [7]. However, this rate is presumed to be heavily underestimated [8].

Over recent years, a wide range of mechanisms related to endometriosis-associated ovarian cancer (EAOC) development have been studied, although the specific mechanisms involved have been not deeply elucidated. It is well known that persistent stimulation by estrogen, without the protection of progesterone, is one of the highest risk factors associated with the malignant transformation of endometriosis [9]. 

Currently, ovarian endometriomas generates a proinflammatory microenvironment that may be conducive to the development of ovarian cancer [10]. Given the evidence of subtypes of ovarian cancer associated with endometriosis, some authors have stated that screening, laboratory, and imaging evaluation should be recommended for the early detection of malignant disorders in women with this disease [11]. However, although multiple scores, including sonography, tumor biomarkers and maternal characteristics, have been postulated, the surgical management of patients with malignant endometriosis remains a challenge in gynecologic oncology.

The potential for malignant transformation of endometriomas has been cited as an indication for surgery—even in asymptomatic patients [12]. Few studies have evaluated the effect of treatment of endometriosis on ovarian cancer risk; however, this would have clear implications. Melin et al. performed a case-control study to analyze the impact of hormonal versus surgical treatment of endometriosis on epithelial ovarian cancer risk. They demonstrated that complete surgical removal of endometriosis lesions as well as removing the affected ovary in the case of ovarian endometriosis may substantially decrease the risk of future ovarian cancer [13]. 

Related to the unexpected cancer when performing conservative laparoscopic surgery in women suffering EAOC is the unintended dissemination of malignant cells. Performing a total adnexectomy directly instead of cyst enucleation could prevent a large proportion of intraoperative spillage in patients with suspicion of ovarian malignancies [14]. However, this approach is not the first choice in women of childbearing age. In addition, many unresolved aspects in endometriosis-associated ovarian cancer remain to be addressed. 

This includes the diagnosis and early detection of malignant transformation of endometriosis, the identification of risk factors associated with the development of ovarian cancer and the stratification of women at increased risk. For this reason, the aim of this study is to analyze the factors of women’s age, body mass index, ultrasound features, and tumor biomarkers to predict endometriosis-associated ovarian cancer in a large series of endometriomas and, secondly, to study the surgical treatment performed in this cohort.

## 2. Materials and Methods

### 2.1. Study Design and Cohort Selection

This was an observational and retrospective study conducted in the endometriosis unit at Genecology Service of the Hospital Universitario La Paz (HULP, Madrid, Spain). We collected data from the medical records of all women ultrasound diagnoses of ovarian cyst classified as endometriomas (benign and with risk of malignancy surgically treated) between January 2019 to July 2021.

Inclusion criteria for the study were patients of legal age who signed the informed consent, patients undergoing surgery for symptomatic or suspected OE, and patients with available data in the medical charts. Exclusion criteria were patients with missing data in the clinical records. Eligible women were divided into two groups, according to the final histology as benign endometriomas (ovarian endometriosis: OE, control) and EAOC (case). Finally, the cohort was conformed with 76 eligible women.

This study was performed in accordance with the Declaration of Helsinki regarding studies in human and was approved by the HULP Research Ethics Committee (Ref. PI-3350). The confidentiality and anonymity of the data was guaranteed in every moment of the study protocol.

### 2.2. Variable Collection

The variables extracted of the medical records were:

**Baseline characteristics**: Women’s age (years), body mass index (BMI, kg/m^2^), preoperative symptoms (including dysmenorrhea, dyspareunia, dyschezia, or dysuria), previous treatments used (including cyclic estroprogestins, progestins, progestin intrauterine device, GNRH agonist, and non-steroidal anti-inflammatory drugs), previous surgery (yes/no), reproductive desire (yes/no, since this question may condition the medical treatment), and laboratory data related to tumor biomarkers cancer antigen (CA) 125 (UI/mL), CA 19.9 (UI/mL) and human epididymis protein 4 (He4, UI/mL) were assayed in plasma following the laboratory medical protocols.

**Preoperative sonographic**: This technique was performed by experts following the IOTA criteria [15,16,17]. The different parameters assessed included the location of the cyst lesion (cm), bilateral mass (yes/no), presence of papillary projections (defined as any solid protrusion into a cyst cavity with a height of ≥3 mm), irregularity of the surface of papillary projections (yes/no), presence of solid tissue other than papillary projections, and presence of septa (according to Timmerman et al. [18] as thin strand of tissue running cross the cyst cavity). Gray-scale and Echo-Doppler ultrasound (Mindray DC60, Shenzhen, China) study were also performed and categorized as non-vascularization, minimal vascularization, moderate vascularization and strong vascularization. 

In addition, the suspicion of malignancy (yes/no) was considered according to IOTA criteria by ultrasound variables [18] as papillary projections and larger than 3 mm, irregularity of the cyst wall, presence of septa and high Doppler score defined as a positive Doppler signal.

### 2.3. Statistical Analysis

Data analyses were conducted using the Statistical Package for the Social Science Software (SPSS Statistics, Version 25.0.0; IBM^®^ Corp., Armonk, NY, USA). The distribution of the quantitative variables was tested using the Shapiro–Wilk test. The quantitative variables were shown as means and standard deviations (SD), or median and interquartile range [Q1; Q3], according to the variable distribution. The qualitative variables were shown as relative frequency (%) and sample size. In quantitative variables, the difference between groups was performed using the Mann–Whitney U test and to study the association between groups in qualitative variables we used Fischer’s exact test. A receiver operating characteristic (ROC) analysis was performed to determine the cut-off point of highest sensitivity and specificity for women’s age and the size of the cyst lesion. 

The area under the curve (AUC) and associated *p*-value were extracted. A binary logistic regression model was used to determine the factors associated with the EAOC histology findings. From the model, we extracted the odds ratio (OR), 95% confidence interval (CI) and explanatory variance (R^2^). A *p*-value of less than 0.05 was considered statistically significant.

## 3. Results

### 3.1. Characterization of the Study Population

Among 76 women of the cohort, 77.6% (59) lesions were benign endometriomas (OE), and 22.4% (17) were EAOC (case). Considering histological examination for the EAOC group, 6.6% (5) were endometrioid carcinoma and 15.8% (12) were clear-cell carcinoma. The women in the EAOC group were significantly higher in age than the women in the OE group. In contrast, the BMI was significantly higher in the OE group compared with in the EAOC group (Table 1). We did not detect any association between the group and preoperative symptoms, previous treatment use, previous surgery, or reproductive desire. In addition, no significant difference was detected among the groups regarding biomarkers. 

The cystectomies performed were 51.4% (36), and the adnexectomies were 48.6% (34). 64.8% (35) of the cystectomies were performed in OE group, and 6.3% (1) were in the EAOC group (this woman was a secondary surgery with complete laparoscopic staging for ovarian cancer including hysterectomy, bilateral salpingo-oophorectomy, pelvic and para-aortic lymphadenectomy, after histopathologic confirmation of clear-cell ovarian cancer). While 35.2% (19) of the adnexectomies were conducted in the control group, and 93.8% (15) were in the EAOC group. We observed an association between the type of surgery performed and the groups (χ^2^ = 16.947, *p*-value < 0.001). 

We also showed a significant association between ultrasound suspicion of malignancy and the EAOC group (*p*-value < 0.001). The ROC analysis determined that the cut-off for age with high sensitivity and specificity was 40 years old (AUC = 0.79, *p*-value < 0.001). A second ROC analysis was performed to determine the cut-off for size of the BMI (AUC = 0.42, *p*-value < 0.001). The best sensitivity and specificity indexes were established with the value of 20.0 kg/m^2^.

### 3.2. Preoperative Ultrasound Features

In relation to the sonographic parameters, the EAOC group presented a larger cyst lesion size compared with the OE group. No association was found between groups regarding the presence of a bilateral mass. However, the EAOC group showed association with increased presence of papillary projections, size of papillary projections within the cyst >3 mm, more irregular surface of the mass, higher presence of septa and positive echo-Doppler ultrasound compared with the OE group (Table 2). In addition, the presence of papillary projections and the size of papillary projections within the cyst >3 mm was associated (χ^2^ = 33.15, *p*-value < 0.001). The size of papillary projections more than 3 mm were extracted from the model to avoid collinearity.

The ROC analysis was performed to determine the cut-off for size of the cyst lesion (AUC = 0.72, *p*-value < 0.001). The best sensitivity and specificity indexes were established with the value of 6 cm.

### 3.3. Contribution of Maternal Variables and Ultrasound Features in the Malignancy of Endometriosis

All variables that showed statistical significance in the univariate analysis were entered into the multivariate logistic regression model to detect the contribution of each variable analyzed to the risk of EAOC. The model was conducted to assess the effect of age, BMI, size of the cyst lesion, presence of papillary projections, size of papillary projections within the cyst >3 mm, irregular surface of the mass, presence of septa, and positive echo-Doppler ultrasound on the likelihood of EAOC finding in histology examination. 

The overall model explained 65.9% of the variance and correctly predicted 77.1% of cases. The variables BMI and irregular surface of the mass were not significant. However, the women’s age was associated with EAOC risk. The EAOC increased 1.16 [1.06; 1.27]-times per year of age of the women, and size of the cyst lesion, presence of papillary projections, septa, and positive echo-Doppler ultrasound (Table 3).

## 4. Discussion

Endometriosis can be considered as an entity with malignancy potential because of its progressive and invasive growth pathways, estrogen-dependent growth, recurrence, and a tendency to metastasize [19]. Therefore, medical professionals dealing with endometriosis face problems in the diagnosis, treatment and follow up of patients [20]. In this study, we found that, for women, high age was a risk factor for EAOC. Ultrasound is the main tool to predict the risk of malignant transformation of endometriosis, and tumor biomarker CA 125 was not useful for this propose. These data can help in the prediction and follow-up of this gynecologic complication.

In our experience, we found that 22.4% of the operated endometriomas corresponded to EAOC. The incidence of this entity in our study is significantly higher than in previous publications [6,21]. This difference may result from the inclusion criteria used in our cohort. Other studies included endometriosis in every entity; however, our study recruited only endometriotic cysts, which might lead to the development of malignant change more often than pelvic endometriosis or other forms of the disease. 

Another important point is that the patients selected were those who received surgery as the elected treatment. Endometriotic cysts managed with other therapies were not taken into consideration. This can lead to an initially higher risk of cancer, as larger cysts and with ultrasound malignant signs are more likely to be excised surgically [22]. Moreover, HULP is considered a tertiary hospital, with a specialized endometriosis unit, where complicated cases of endometriosis are referred from other Spanish centers for specific management, and this could lead to a more challenging patient profile. All these factors could explain the higher EAOC incidence in our study.

Although the mean age at onset of ovarian cancer is approximately 56 years [23], the onset of endometriosis occurs mostly during women’s reproductive age. The reported mean age of EAOC cases is often significantly younger than ovarian cancer patients without endometriosis but older than women with endometriosis alone [24]. In agreement with this, our study demonstrates that the age of women in the EAOC group was significantly higher than in the benign endometriosis (ovarian endometriosis) group.

Regarding the histological subtypes of ovarian cancer in our study, clear-cell carcinoma was more frequent than endometrioid carcinoma. We found out that 15.8% of the cases of cancer were clear-cell carcinoma, and 6.6% were endometrioid carcinoma. Similar results were shown in previous studies [25]. However, other studies showed equal rates between clear-cell and endometrioid subtypes (39% versus 35%) [24]. We did not find any case of serous carcinoma, which is the most common form of epithelial ovarian cancer. This finding was consistent with previous data, where the incidence of serous carcinoma was lower than clear-cell and endometrioid forms [24]. This evidence supports the fact that ovarian cancer associated with endometriosis has unique characteristics.

In addition, this article demonstrates that age of the patients and size of the cyst lesion have significant associations with the EAOC group; therefore, these may be important risk factors to consider during the preoperatory assessment of these women. Similarly, Udomsinkul et al. retrospectively evaluated 79 cases of confirmed endometriosis-associated ovarian cancer and showed that advanced age, menopause, weight loss before treatment, larger cysts, and complex cysts (solid component) were important and independent risk factors for EAOC [25]. Moreover, Thomsen et al. conducted a systematic review and concluded that older age at endometriosis diagnosis (≥45 years), nulliparity, hyperestrogenism, solid compartments, as well as larger size of endometrioma (≥9 cm in diameter) were associated with an increased risk of ovarian cancer among women with endometriosis [26]. 

Following the same line of research, a retrospective study performed in a Taiwanese population identified demographic and clinical factors, such as increased age, living in an urbanized area, low or high income, depression, pelvic inflammation, and the absence of childbearing post-endometriosis as risk factors for developing ovarian cancer in patients with endometriosis [27].

Regarding tumor biomarkers, it is well known that serum marker of CA 125 is a gold standard to monitor the therapeutic response and detection of recurrent disease in patients with epithelial ovarian cancer [28]. Therefore, biomarker CA 125 evaluation may be helpful when considering the risk of malignancy transformation [29]. In this context, Wang et al. compared the clinical and biologic characteristics between women with endometriosis-associated ovarian cancer and women with epithelial ovarian cancer non associated with endometriosis and proved that patients in the first group had lower preoperative serum level of CA 125 (122.9 versus 1377.5 U/mL) and were more likely to display normal CA 125 levels [30]. 

This evidence could suggest that the entity of ovarian cancer related to endometriosis does not significantly rise the levels of this tumor biomarker and this is in concordance with our results, where CA 125 did not show a statistical difference between the EAOC group and the benign endometriomas (ovarian endometriosis) group. However, in this article, the percentage of missing data regarding this variable was high (55.3% in the first group and 30.8% in the second group), and this could affect the statistical analysis. 

In contrast with our results, a recent study compared the levels of tumor biomarkers He4 and CA 125 between patients with EAOC, patients with benign endometriomas and a group of healthy control woman and proved that He4 and CA125 expression levels were significantly higher in the EAOC group than in the other two groups [31]. It is important to note that the tumor biomarkers could be useful to predict EAOC. However, we recommend requesting in well-defined context, such as patients with ultrasound suspicion of malignancy, with an increase in the size of the cyst or change in the ultrasound image, and in perimenopausal women with new onset pelvic pain.

As is well known, EAOC is a challenging entity both in terms of early diagnosis and therapeutic management related to sonographic differentiation and surgical management needs to be assessed in the preoperatory time. Regarding the therapy, benign endometrioma are usually managed surgically to preserve the ovary. In contrast, EAOC must be managed surgically by complete salpingo-oophorectomy without opening the cystic lesions to avoid intra-abdominal spillage of malignant cells [32]. In our experience, the percentage of adnexectomies performed in the EAOC group was 93.8%. This suggests that a satisfactory surgical planification was performed. 

These data highlight the importance of managing these cases in high specialized level hospital. On the other side, EAOC early diagnosis requires a high degree of expertise in transvaginal sonography, which is the most useful and accessible method in preoperative assessment. Clinical research collaborations, such as the International Ovarian Tumor Analysis (IOTA) group, have considerably improved the ultrasound assessment of suspicious ovarian masses [33]. In our study, we show a significant association between ultrasound suspicious of cancer and EAOC diagnosis. 

Timmerman et al. prospectively assessed the diagnostic by ultrasound rules to predict benignity/malignancy in an adnexal mass and found the use of this rule yielded a conclusive result in 77.0% of masses (sensitivity = 92% and specificity = 96%). Malignant features include an irregular solid tumor, ascites, at least four papillary projections, irregular multilocular solid tumor with the largest diameter ≥10 cm, and strong intratumoral blood flow on Doppler [16]. Alcázar et al. assessed the diagnostic using a three-step strategy proposed by IOTA for discriminating between benign and malignant adnexal masses and confirmed the value of this ultrasound rule with an overall sensitivity of 94.3% and specificity of 94.9% [17]. 

In our study, we used these rules to characterize the ultrasound reports performed by experts and we labelled them as suspicious of malignancy or non-suspicious of malignancy according to the presence or absence of malignant features. In our experience, the EAOC group showed association with increased presence of papillary projections, size of papillary projections within the cyst higher than 3 mm, more irregular surface of the mass, higher presence of septa and positive echo-Doppler ultrasound compared with the ovarian endometriosis group. Moreover, we showed a significant association between the ultrasound suspicion of cancer and EAOC diagnosis. This data can support the use of IOTA rules for the accurate diagnosis of malignant endometriosis.

### Strength and Limitations

Our paper is consistent with previous data and current evidence regarding the impact of preoperatory factors in the surgical assessment of endometriomas. Age of the patients and size of the cyst play an important role when considering the risk of malignancy associated with endometriosis. Moreover, we highlight the importance of detailed ultrasound characterization of the masses following IOTA criteria assessment. Therefore, most interesting in our study was that, in addition to relating EAOC to preoperative clinical and sonographic factors, we related these risk factors for malignancy to the appropriate surgery (adnexectomy versus cystectomy). 

However, we are aware of the limitations of a retrospective design, such as missing data and selection bias, which requires further research in this field. Although, considering the overall prevalence of malignant endometriosis-associated ovarian cancer, we achieved a large sample size for the EAOC group, it should be considered that the models were unadjusted, and further research would be necessary to corroborate the risk of these variables in the correct surgical management of these patients, i.e., considering to the histological types and tumor markers, each EAOC women will be different. Therefore, an individual analysis could be required.

Currently, some lines of study are focusing on immunotherapy as an innovative therapeutic option for the EAOC. A recent study performed by Nero et al. demonstrated higher levels of PD-1/PD-L1 expression in endometriosis-associated ovarian cancer compared with benign endometriosis-related diseases, such as ovarian endometriosis and atypical endometriosis [34]. Interestingly, several clinical trials using monoclonal antibodies targeting PD-1 and PD-L1 in patients with EAOC are in progress [35].

## 5. Conclusions

In our experience, the age of the patient and preoperatory ultrasound features, such as papillary projections, septa, and positive echo-Doppler, were the main factors to consider when evaluating the malignancy risk associated with ovarian endometriosis, particularly in women older than 40 years. Furthermore, we recommend considering the relationship of the papillary projection, the size of the cyst (cut-off higher than 6 cm), and size of papillary projection higher than 3 mm. However, the tumor biomarkers were not as predictive in the management of endometriosis-associated ovarian cancer. These data might help to create a pre-operatory algorithm to guide the therapeutic management and surgical decision-making of these patients. Further studies need to be done following this line of investigation.

## Figures and Tables

**Table 1 jcm-11-02414-t001:** Baseline clinical characteristics between groups.

	OE (*n* = 59)	EAOC (*n* = 17)	*p*-Value
Women’s age (years)	39.0 [34.0; 46.0]	50.0 [43.0; 63.0]	<0.001 ^a^
Body mass index (kg/m2)	24.9 (5.3)	23.3 (4.6)	<0.001 ^b^
Preoperative symptoms	96.6% (57)	100% (17)	0.736 ^c^
Previous treatments used	100% (59)	64.7% (11)	0.461 ^c^
Previous surgery	64.4% (38)	100% (17)	0.373 ^c^
Reproductive desire	100% (59)	41.2% (7)	0.999 ^c^
CA 125 (UI/mL)	43.8 [22.8; 111.8]	40.9 [12.7; 165.0]	0.517 ^a^
CA 19.9 (UI/mL)	41.4 [18.0; 78.2]	17.5 [4.4; 1161.7]	0.612 ^a^
He4 (UI/mL)	57.3 [36.9; 115.6]	411.0 [78.0; 1083.0]	0.143 ^a^

In quantitative variables, the data show median and interquartile range [Q; Q3] for non-normally distributed variables and mean and standard deviation (SD) for normally distributed variables. Qualitative variables are described as the relative frequency and sample size (*n*). Cancer antigen (CA), Human epididymis protein 4 (He4). The *p*-value was extracted from the ^a^ Mann–Whitney U test, ^b^ Student’s *t*-test or ^c^ Fischer’s exact test. OE: Ovarian endometriosis; EAOC: Endometriosis-associated ovarian cancer.

**Table 2 jcm-11-02414-t002:** Sonographic feature outcomes between groups.

	OE (*n* = 59)	EAOC (*n* = 17)	*p*-Value
Size of the cyst lesion (cm)	6.0 [4.0; 8.0]	7.6 [6.0; 12.0]	0.024 ^a^
Bilateral mass	55.9% (33)	64.7% (11)	0.519 ^b^
Presence of papillary projections	11.9% (7)	82.4% (14)	<0.001 ^b^
Size of papillary projections within the cyst >3 mm	0.0% (0)	64.7% (11)	<0.001 ^b^
Irregular surface of the mass	0.0% (0)	70.6% (12)	<0.001 ^b^
Presence of septa	3.4% (2)	70.6% (12)	<0.001 ^b^
Positive echo-Doppler ultrasound	6.8% (4)	88.2% (15)	<0.001 ^b^

In quantitative variables, the data show the median and interquartile range [Q; Q3]. In qualitative variables, the data are described as the relative frequency and sample size (n). The *p*-value was extracted from the ^a^ Mann–Whitney U test and ^b^ Fischer’s exact test. OE: Ovarian endometriosis; EAOC: Endometriosis-associated ovarian cancer.

**Table 3 jcm-11-02414-t003:** Multivariate logistic regression models associated with the malignancy endometriosis risk.

	β	SE	OR [95% CI]	*p*-Value
Women’s age	0.149	0.045	1.16 [1.06; 1.27]	0.001
Body mass index	−0.069	0.061	0.93 [0.83; 1.05]	0.258
Size of the cyst lesion	0.169	0.070	1.18 [1.03; 1.36]	0.016
Presence of papillary projections	3.526	0.753	34.0 [7.77; 148.8]	<0.001
Irregular surface of the mass	23.64	11,602.7	1.8 × 10^10^ [0.00; ∞]	0.999
Presence of septa	4.190	0.895	66.0 [11.4; 381.6]	<0.001
Positive echo-Doppler ultrasound	5.292	1.156	198.8 [20.6; 1914.3]	<0.001

Data shown beta coefficients (β), standard error (SE), odds ratio (OR) and 95% confidence interval (CI). The *p*-value of each factor was extracted.

## Data Availability

The data presented in this study are available on request from the corresponding author.

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
