# Peer review of "Evaluation of Women’s Age and Ultrasonographic Features to Choose Surgical Treatment for Endometriosis Associated with Ovarian Cancer"

_jcm, 2022, doi:10.3390/jcm11092414_

Round 1
Reviewer 1 Report
Comments:
- There is one mistake in Abstract. In line 32-33, the sentence “The age of women in non-EAOC group was 39.0 [34.0; 46.0] years being significantly higher in EAOC group (50.o [43.0; 63.0] years; p-Value < 0.001)” is mistake or confused to the readers. According to the results in the study, women in EAOC group have higher age than women in non-EAOC group.
- There is a contradiction in the results. The authors concluded that age, preoperatory ultrasound features, such as size of endometrioma, presences of papillary projections, septa, irregular surface of the mass and positive echo-Doppler, were the main risk factors for malignant transformation (Table 2). However, further analysis by multivariate logistic regression models associated to malignancy endometriosis risk revealed that presence of papillary projections, presence of septa and positive echo-Doppler ultrasound have lower odds ratio associated with malignancy risk (Table 3). The authors should further explain it in Discussion.
- The title of the table 2 should be modified.
Author Response
Response: The authors thank the time dedicated to the article and the comments provided, which we believe have substantially improved the understanding of the manuscript.
There is one mistake in Abstract. In line 32-33, the sentence “The age of women in non-EAOC group was 39.0 [34.0; 46.0] years being significantly higher in EAOC group (50.0 [43.0; 63.0] years; p-Value < 0.001)” is mistake or confused to the readers. According to the results in the study, women in EAOC group have higher age than women in non-EAOC group.
Response: Totally agree and excuse us by this error which has been corrected in the revised version.
There is a contradiction in the results. The authors concluded that age, preoperatory ultrasound features, such as size of endometrioma, presences of papillary projections, septa, irregular surface of the mass and positive echo-Doppler, were the main risk factors for malignant transformation (Table 2). However, further analysis by multivariate logistic regression models associated to malignancy endometriosis risk revealed that presence of papillary projections, presence of septa and positive echo-Doppler ultrasound have lower odds ratio associated with malignancy risk (Table 3). The authors should further explain it in Discussion.
Response: The authors apologize for the transcription error. We have deeply reviewed the analysis and have made the corrections.
The title of the table 2 should be modified.
Response: Following the reviewer's recommendations, the title of Table 2 has been modified.
Reviewer 2 Report
This is an intersting study on an important topic. The sample size is limited, the study is limited by its retrospective design, but these have been addressed in the limitations.
Author Response
This is an interesting study on an important topic. The sample size is limited, the study is limited by its retrospective design, but these have been addressed in the limitations.
Response: The authors thank the time dedicated to the article and the comments provided. We fully agree with this comment. However, although the size is limited, the prevalence of EAOC in endometriosis is very low and, in this study, we have obtained a considerable group of women with this complication. We have clarified this point in the discussion section (lines 328-333).
Reviewer 3 Report
It is considered a very meaningful study to analyze the factors that can predict endometriosis and accompanying ovarian cancer. However, there are some limitations that do not fit the purpose of this journal.
First, out of 76 patients with endometriosis, only 17 had ovarian cancer. There will be various histological types among ovarian cancer patients, and the shape and tumor markers for each will be different, so an individual analysis will be required. However, the number of subjects is so small that the analysis seems to be limited.
Second, the topic is not new or meaningful. It is well known that ovarian cancer is associated with the age of the patient. In addition, the predictive power of IOTA criteria for ovarian cancer has already been well proven. If you want to give meaning only to endometriosis, an additional number of subjects should be recruited.
Author Response
It is considered a very meaningful study to analyze the factors that can predict endometriosis and accompanying ovarian cancer. However, there are some limitations that do not fit the purpose of this journal.
Response: The authors thank the time dedicated to the article and the comments provided, which we believe have substantially improved the understanding of the manuscript. We wanted to include this article in the Special Issue “Endometriosis: Current Perspectives on Diagnosis and Treatment”. Therefore, we thought it might be interesting to report this data.
First, out of 76 patients with endometriosis, only 17 had ovarian cancer. There will be various histological types among ovarian cancer patients, and the shape and tumor markers for each will be different, so an individual analysis will be required. However, the number of subjects is so small that the analysis seems to be limited.
Response: We fully agree with this comment. EAOC is very aggressive cancer with low prevalence in patients with endometriosis. We could enroll 17 women with EAOC. In other hand, in early stage of cancer, tumor markers could have low sensitivity increasing in conditions such as endometriosis and smoking (PMID: 30917847). Furthermore, recent recommendations report to not follow-up serum markers in women with unsuspicious ovarian endometriosis (PMID: 31206037). According to our data, the tumor marker may not be indicative of malignancy at recovery, but the ultrasound features are. However, we have included this point of individuality in the limitations of the study.
Second, the topic is not new or meaningful. It is well known that ovarian cancer is associated with the age of the patient. In addition, the predictive power of IOTA criteria for ovarian cancer has already been well proven. If you want to give meaning only to endometriosis, an additional number of subjects should be recruited.
Response: We are fully agreeing with the reviewer’s comment. However, the novelty of this manuscript was to focus on low-prevalent type of ovarian cancer (EAOC), and to explore the criteria that a gynecologist must consider and that can condition surgery. Our data would serve to support the IOTA criteria in the Spanish context, i.e., other study from Thai stablished a cut-off for cyst diameter in 8 cm (PMID: 32127149), while we could stablish 6 cm.
Round 2
Reviewer 1 Report
1. The authors concluded that age of the patient and preoperatory ultrasound feature, such as papillary projections, septa, and positive echo-Doppler, were the main factors to consider when evaluating the malignancy risk associated to ovarian endometriosis. How about the cutoff values, sensitivity and specificity in ROC analysis?
Author Response
The authors concluded that age of the patient and preoperatory ultrasound feature, such as papillary projections, septa, and positive echo-Doppler, were the main factors to consider when evaluating the malignancy risk associated to ovarian endometriosis. How about the cutoff values, sensitivity, and specificity in ROC analysis?
Response: Thank you to review our manuscript. Considering to the reviewer’s comment, we have included the cut-off values for age and size of the cyst lesion in the conclusions. However, the cut-off for BMI was not incorporated due to was not significant in our models and our conclusions were not derived from this factor.
Reviewer 3 Report
In the previous papers (Differentiation between endometriosis-associated ovarian cancers and non- endometriosis-associated ovarian cancers based on magnetic resonance imaging, 2021), “non-EAOC” was defined as ovary cancer not associated with endometriosis, but you defined it as benign endometriosis. This will cause confusion for readers including me. Terminology needs correction.
#1 abstract
Line 35, “tumor biomarkers levels were significantly different”? It is different from the results content in the main body.
#2 Table 1.
In EOAC group, CA125 : 40.9[12.7; 165.0], CA19-9 17.5[4.4; 1161.7], HE4 411.0[78.0; 1083.0] were shown. The range of fluctuation of tumor biomarkers in EOAC is large. Nevertheless, can it be said that tumor markers are meaningless in predicting EOAC?
Author Response
In the previous papers (Differentiation between endometriosis-associated ovarian cancers and non- endometriosis-associated ovarian cancers based on magnetic resonance imaging, 2021), “non-EAOC” was defined as ovary cancer not associated with endometriosis, but you defined it as benign endometriosis. This will cause confusion for readers including me. Terminology needs correction.
Response: We appreciate the time spent in our article. We fully agree with this comment, and we have modified the terminology of non-EAOC to avoid misunderstanding as OE (ovarian endometriosis).
#1 abstract. Line 35, “tumor biomarkers levels were significantly different”? It is different from the results content in the main body.
Response: Previously, the sentences expressed “However, any of the tumor biomarkers levels were significantly different among the groups”. We have modified the sentences to avoid confusions to: “However, tumor biomarkers levels were not significantly different among the groups”.
#2 Table 1. In EOAC group, CA 125: 40.9 [12.7; 165.0], CA19-9 17.5 [4.4; 1161.7], HE4: 411.0 [78.0; 1083.0] were shown. The range of fluctuation of tumor biomarkers in EOAC is large. Nevertheless, can it be said that tumor markers are meaningless in predicting EOAC?
Response: This is a great point. The tumor markers could be useful to predict EAOC. However, we recommend requesting it in well-defined context, such as patients with ultrasound suspicion of malignancy, with an increase in the size of the cyst or change in the ultrasound image, and in perimenopausal women with new onset pelvic pain. To clarify this point, we have included this information in the article.